# Deep learning model calibration for improving performance in class-imbalanced medical image classification tasks

**Sivaramakrishnan Rajaraman**[1]*, **Prasanth Ganesan**[2], **Sameer Antani**[1]

**1** National Library of Medicine, National Institutes of Health, Bethesda, MD, United States of America,
**2** Stanford University Department of Medicine, Stanford, CA, United States of America

* sivaramakrishnan.rajaraman@nih.gov

## Abstract

In medical image classification tasks, it is common to find that the number of normal samples far exceeds the number of abnormal samples. In such class-imbalanced situations, reliable training of deep neural networks continues to be a major challenge, therefore biasing the predicted class probabilities toward the majority class. Calibration has been proposed to alleviate some of these effects. However, there is insufficient analysis explaining whether and when calibrating a model would be beneficial. In this study, we perform a systematic analysis of the effect of model calibration on its performance on two medical image modalities, namely, chest X-rays and fundus images, using various deep learning classifier backbones. For this, we study the following variations: (i) the degree of imbalances in the dataset used for training; (ii) calibration methods; and (iii) two classification thresholds, namely, default threshold of 0.5, and optimal threshold from precision-recall (PR) curves. Our results indicate that at the default classification threshold of 0.5, the performance achieved through calibration is significantly superior ($p < 0.05$) to using uncalibrated probabilities. However, at the PR-guided threshold, these gains are not significantly different ($p > 0.05$). This observation holds for both image modalities and at varying degrees of imbalance. The code is available at https://github.com/sivaramakrishnan-rajaraman/Model_calibration.

## Introduction

Deep learning (DL) methods have demonstrated incredible gains in the performance of computer vision processes such as object detection, segmentation, and classification, which has led to significant advances in innovative applications [1]. DL-based computer-aided diagnostic systems have been used for analyzing medical images as they provide valuable information about the disease pathology. Some examples include chest X-rays (CXRs) [2], computed tomography (CT), magnetic resonance (MR), fundus images [3], cervix images [4], and ultrasound echocardiography [5], among others. Such analyses help in identifying and classifying disease patterns, localizing and measuring disease manifestations, and recommending therapies based on the predicted stage of the disease.

**Funding:** This study is supported by the Intramural Research Program (IRP) of the National Library of

Medicine (NLM) and the National Institutes of Health (NIH). The intramural research scientists (authors) at the NIH dictated study design, data collection, data analysis, decision to publish and preparation of the manuscript.

**Competing interests:** The authors have declared that no competing interests exist.

The success of DL models is due to not only the network architecture but significantly due to the availability of large amounts of data for training the algorithms. In medical applications, we commonly observe that there is a high imbalance between normal (no disease finding) and abnormal data. Such imbalance is undesirable for training DL models. The bias introduced by class-imbalanced training is commonly addressed by tuning the class weights [6]. This step attempts to compensate for the imbalance by penalizing the majority class. However, this does not eliminate bias. Improvements in the accuracy of the minority class achieved through changes in class weights occur at the cost of reducing the performance of the majority class. Data augmentation [7] and random under-sampling [8] are other widely followed techniques for handling class imbalance that has demonstrated performance improvement in several studies. However, in scenarios where augmentation may adversely distort the data characteristics, model calibration may be explored for compensating for the imbalance.

Model calibration refers to the process of rescaling the predicted probabilities to make them faithfully represent the true likelihood of occurrences of classes present in the training data [9]. In healthcare applications, the models are expected to be accurate and reliable. Controlling classifier confidence helps in establishing decision trustworthiness [10]. Several calibration methods have been proposed in the literature including Platt scaling, isotonic regression, beta calibration, spline calibration, among others [11–13]. A recent study used calibration methods to rescale the predicted probabilities toward text and image processing tasks [9]. The authors observed that the DL models trained with batch normalization layers demonstrated higher miscalibration. It was also observed that the calibration was negatively impacted while training with reduced weight decay. Another study [14] experimented with ImageNet, MNIST, Fashion MNIST, and other natural image datasets to analyze calibration performance through the use of adaptive probability binning strategies. They demonstrated that calibrated probabilities may or may not improve performance and it depends on the performance metric used to assess predictions. The authors of [15] used AlexNet [16], ResNet-50 [17], DenseNet-121 [18], and SqueezeNet [19] models as feature extractors to extract and classify features from four medical image datasets. The predicted probabilities were rescaled and mapped to their true likelihood of occurrence using a single-parameter version of Platt scaling. It was observed that the expected calibration error (ECE) decreased by 65.72% compared to that obtained with their uncalibrated counterparts while maintaining classification accuracy. In another study [20], the authors used the single-parameter version of Platt scaling to calibrate the prediction probabilities toward a multi-class polyp classification task. It was observed that the ECE and maximum calibration error (MCE) were reduced using calibrated probabilities and resulted in improved model interpretability. The authors of [21] used the single-parameter version of Platt scaling to calibrate probabilities obtained toward an immunofluorescence classification task using renal biopsy images. It was observed that the ECE values reduced after calibration, however, it resulted in reduced accuracy, compared to their uncalibrated counterparts. These studies establish that calibration reduces errors due to the mismatch between the predicted probabilities and the true likelihood of occurrence of the events. However, the literature lacks a detailed analysis of the relationship between the degree of data imbalance, the calibration methods, and the effect of the classification threshold on model performance before and after calibration.

Our novel contribution is a study of class-imbalanced medical image classification tasks that investigates: (i) selection of calibration methods for superior performance; (ii) finding an optimal "calibration-guided" threshold for varying degrees of data imbalances, and (iii) statistical significance of performance gains through the use of a threshold derived from calibrated probabilities over default classification threshold of 0.5. Accordingly, we evaluate the model performance before and after calibration using two medical image modalities, namely, CXRs

and fundus images. We used the Shenzhen TB CXRs [22] dataset and the fundus images made available by the Asia Pacific Tele-Ophthalmology Society (APTOS) to detect diabetic retinopathy (DR). Next, we artificially vary the degrees of data imbalance in the training dataset such that the abnormal samples are 20%, 40%, 60%, 80%, and 100% proportions of normal samples. We investigate the performance of several DL models, namely, VGG-16 [23], Densenet-121 [18], Inception-V3 [24], and EfficientNet-B0 [25], which have been shown to deliver superior performance in medical computer vision tasks. We evaluated the impact on the performance using three calibration methods, namely, Platt scaling, beta calibration, and spline calibration. Each calibration method is evaluated using the ECE metric. Finally, we studied the effect of two classification thresholds. One is the default classification threshold of 0.5, and the other is the optimal threshold derived from the precision-recall (PR) curves. The performance with calibrated probabilities is compared to that obtained using the uncalibrated probabilities for both the default classification threshold (0.5) and PR-guided optimal classification threshold.

## Materials and methods

### Dataset characteristics

The following datasets are used in this retrospective study:

i. APTOS'19 fundus: A large-scale collection of fundus images obtained through fundus photography are made publicly available by the Asia Pacific Tele-Ophthalmology Society (APTOS) for the APTOS'19 Blindness Detection challenge (https://www.kaggle.com/c/aptos2019-blindness-detection/overview). The goal of the challenge is to classify them as showing normal retina or signs of diabetic retinopathy (DR). Those showing signs of DR are further categorized on a scale of 0 (no DR) to 4 (proliferative DR) based on disease severity. Variability is introduced into the data by gathering them from multiple sites at varying periods using different types of cameras. In our study, we took 1200 fundus images showing normal retina and a collection of 1200 images showing a range of disease severity, i.e., 300 images each from each severity level 1–4.

ii. Shenzhen TB CXR: A set of 326 CXRs showing normal lungs and 336 CXRs showing other Tuberculosis (TB)-related manifestations were collected from the patients at the No.3 hospital in Shenzhen, China. The dataset was de-identified, exempted from IRB review (OHSRP#5357), and released by the National Library of Medicine (NLM). An equal number of 326 CXRs showing normal lungs and TB-related manifestations are used in this study. All images are (i) resized to 256×256 spatial resolution, (ii) contrast-enhanced using Contrast Limited Adaptive Histogram Equalization (CLAHE) algorithm, and (iii) rescaled to the range [0 1] to improve model stability and performance.

### Simulating imbalance in the training dataset

The datasets are further divided into multiple sets with varying degrees of imbalance of the positive disease samples. The sets are labeled as Set-N, where N is one of {20, 40, 60, 80, 100} and represents the proportion of disease-positive samples to disease-negative samples. Therefore, Set-100 has an equal number of disease-positive and disease-negative samples. For reasons of brevity, and because the results demonstrate a similar trend, in the remainder of this manuscript, we present results from only Set-20, Set-60, and Set-100. For completeness, we provide results from Set-40 and Set-80 as supplementary materials. The number of images in the train and test set for each of these datasets is shown in Table 1.

**Table 1. Class imbalance-simulated sets constructed from the datasets used in this study.**

| Data | Shenzhen TB CXR | | | | APTOS'19 fundus | | | |
|---|---|---|---|---|---|---|---|---|
| | Train | | Test | | Train | | Test | |
| | No finding | TB | No finding | TB | No finding | DR | No finding | DR |
| Set-100 | 226 | 226 | 100 | 100 | 1000 | 1000 | 300 | 300 |
| Set-80 | 226 | 180 | 100 | 100 | 1000 | 800 | 300 | 300 |
| Set-60 | 226 | 136 | 100 | 100 | 1000 | 600 | 300 | 300 |
| Set-40 | 226 | 90 | 100 | 100 | 1000 | 400 | 300 | 300 |
| Set-20 | 226 | 45 | 100 | 100 | 1000 | 200 | 300 | 300 |

## Classification models

We used four popular and high-performing DL models in this study, namely, VGG-16, Dense-Net-121, Inception-V3, and EfficientNet-B0. These models have demonstrated superior performance in medical computer vision tasks [1]. These models are (i) instantiated with their ImageNet-pretrained weights, (ii) truncated at their deepest convolutional layer, and (iii) appended with a global average pooling (GAP) layer, a final dense layer with two output nodes and Softmax activation to output class predictions.

First, we selected the DL model that delivered a superior performance with the Shenzhen TB CXRs and the APTOS'19 fundus datasets. In this regard, the models are retrained on the Set-100 dataset from the (i) Shenzhen TB CXR and (ii) APTOS'19 fundus datasets to predict probabilities toward classifying them to their respective categories. Of the number of training samples in the Set-100 dataset, 10% of the data is allocated to validation with a fixed seed. We used a stochastic gradient descent optimizer with an initial learning rate of 1e-4 and a momentum of 0.9. Callbacks are used to store model checkpoints. The learning rate is reduced whenever the validation loss plateaued. The weights that delivered a superior performance with the validation set are further used for predicting the test set.

The best-performing model with the balanced Set-100 dataset is selected for further analysis. We instantiated the best-performing model with their ImageNet-pretrained weights, added the classification layers, and retrained it on the Set-20 and Set-60 datasets that are constructed individually from the (i) Shenzhen TB CXR and (ii) APTOS'19 fundus datasets to record the performance. Fig 1 shows the general block diagram with various dataset inputs to the DL models and their corresponding dataset-specific predictions.

## Evaluation metrics

The following metrics are used to evaluate the models' performance: (a) Accuracy, (b) area under the precision-recall curve (AUPRC), (c) F-score, and (d) Matthews correlation coefficient (MCC). These measures are expressed as shown below:

$$Accuracy = \frac{TP + TN}{TP + TN + FP + FN} \tag{1}$$

$$Recall = \frac{TP}{TP + FN} \tag{2}$$

$$Precision = \frac{TP}{TP + FP} \tag{3}$$

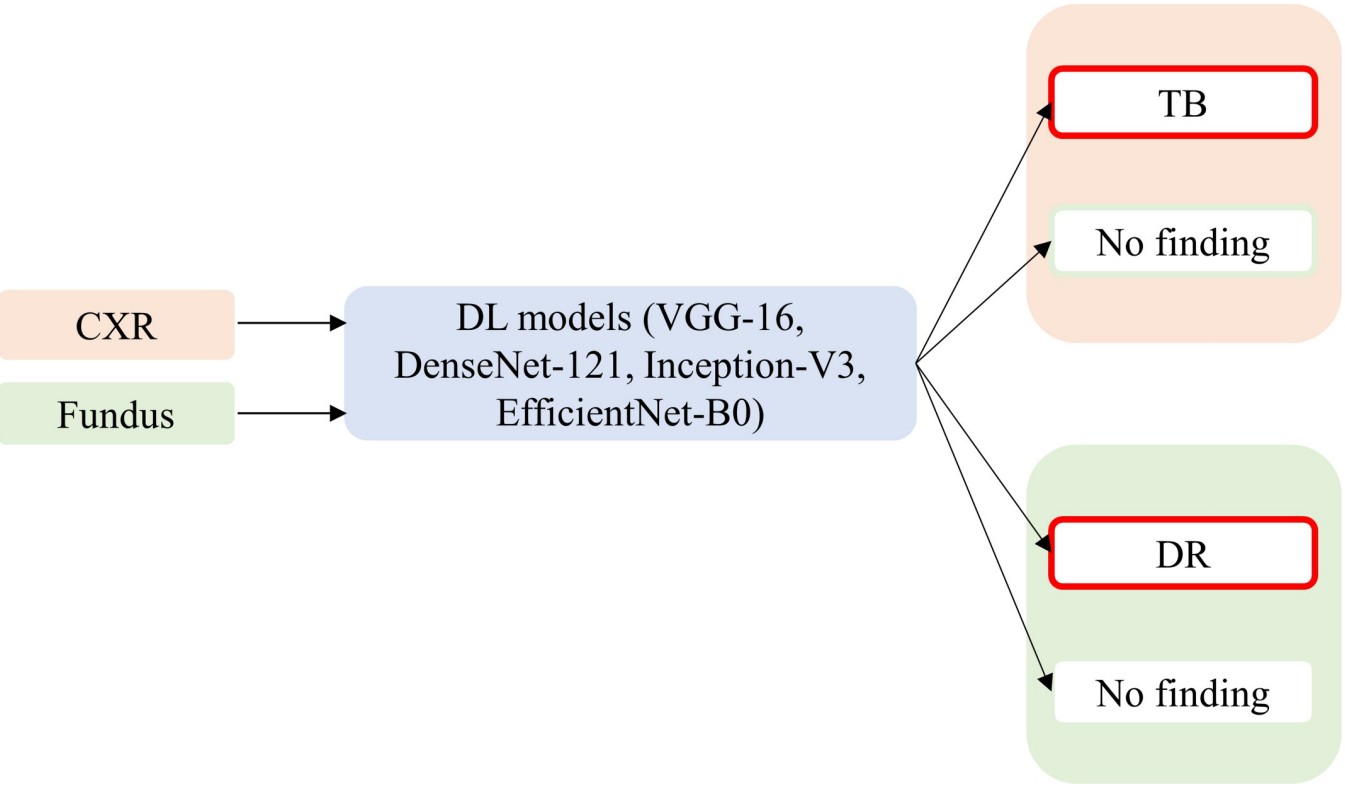

**Fig 1. Block diagram showing the various dataset inputs to the DL models and their corresponding dataset-specific predictions.**

$$F-score = 2 \times \frac{Precision \times Recall}{Precision + Recall} \tag{4}$$

$$MCC = \frac{TP \times TN - FP \times FN}{((TP+FP)(TP+FN)(TN+FP)(TN+FN))^{1/2}} \tag{5}$$

Here, TP, TN, FP, and FN denote the true positive, true negative, false positive, and false negative values, respectively. We used Tensorflow Keras version 2.4 and CUDA dependencies to train and evaluate the models in a Windows$^{®}$ computer with Intel Xeon processor and NVIDIA GeForce GTX 1070 GPU.

### Threshold selection

The evaluation is first carried out using the default classification threshold of 0.5, i.e., predictions $>=0.5$ will be categorized as abnormal (disease-class) and those that are $<0.5$ will be categorized as samples showing no findings. However, using a theoretical classification threshold of 0.5 may adversely impact classification particularly in an imbalanced training scenario [26]. The study in [27] reveals that it would be misleading to resort to data resampling techniques without trying to find the optimal classification threshold for the task. There are several approaches to finding the optimal threshold for the classification task. These are broadly classified into (i) ROC curve-based methods [28, 29] and (ii) Precision-recall (PR) curve-based methods [30]. In ROC curve-based approach, different values of thresholds are used to

interpret the false-positive rate (FPR) and true-positive rate (TPR). The area under the ROC curve (AUROC) summarizes the model performance. A higher value for the AUROC (close to 1.0) signifies superior performance. Metrics such as geometric means (G-means) and Youden statistic (J) are evaluated to identify this optimal threshold from ROC curves. The optimal threshold results in a superior balance of precision and recall and can be measured using the PR curve. The value of the F-score is computed for each threshold and its largest value and the corresponding threshold are recorded. This threshold is then used to predict test samples and convert the class probabilities to crisp image-level labels. Unlike ROC curves, the PR curves focus on model performance for the positive disease class that is the high-impact event in a classification task. Hence, they are more informative than the ROC curves, particularly in an imbalanced classification task [30]. Thus, we selected the optimal threshold from the PR curves.

## Calibration: Definition

The goal of calibration is to find a function that fits the relationship between the predicted probability and the true likelihood of occurrence of the event of interest. Let the output of a DL model $D$ be denoted by $h(D) = (X', P')$, where $X'$ is the class label obtained from the predicted probability $P'$ that needs to be calibrated. If the outputs of the model are perfectly calibrated then,

$$\mathbb{P}(X' = X | P' = p) = p, \forall p \in [0, 1] \tag{6}$$

## Qualitative evaluation of calibration—reliability diagram

The reliability diagram, also called the calibration curve, provides a qualitative description of calibration. It is plotted by dividing the predicted probabilities into a fixed number of bins $Z$, each of size $1/Z$, and having equal width, along the x-axis. Let $C_z$ denote the set of sample indices whose predicted probabilities fall into the interval $I_z = \left( \frac{z-1}{Z}, \frac{z}{Z} \right)$, for z ∈ {1, 2, . . ., Z}. The accuracy of the bin $C_z$ is given by,

$$Accuracy\ (C_z) = 1/|C_z| \sum_{i \in C_z} 1(y_i' = y_i) \tag{7}$$

The average probability in the bin $C_z$ is given by:

$$Average\ Probability\ (C_z) = 1/|C_z| \sum_{i \in C_z} p_i' \tag{8}$$

Here, $p_i'$ is the predicted probability for the sample $i$. With improving calibration, the points will lie closer to the main diagonal that extends from the bottom left to the top right of the reliability diagram. Fig 2 shows a sample sketch of the reliability diagram. The points below the diagonal indicate that the model is overconfident, and the predicted probabilities are too large. Those above the diagonal indicate that the model is underconfident, and the predicted probabilities are too small.

## Quantitative evaluation of calibration: Expected calibration error (ECE)

The ECE metric provides a quantitative measure of miscalibration. It is given by the expectation difference between the predicted probabilities and accuracy as shown below:

$$ECE = \sum_{z=1}^{Z} \frac{|C_z|}{m} \left| accuracy\ (C_z) - probability\ (C_z) \right| \tag{9}$$

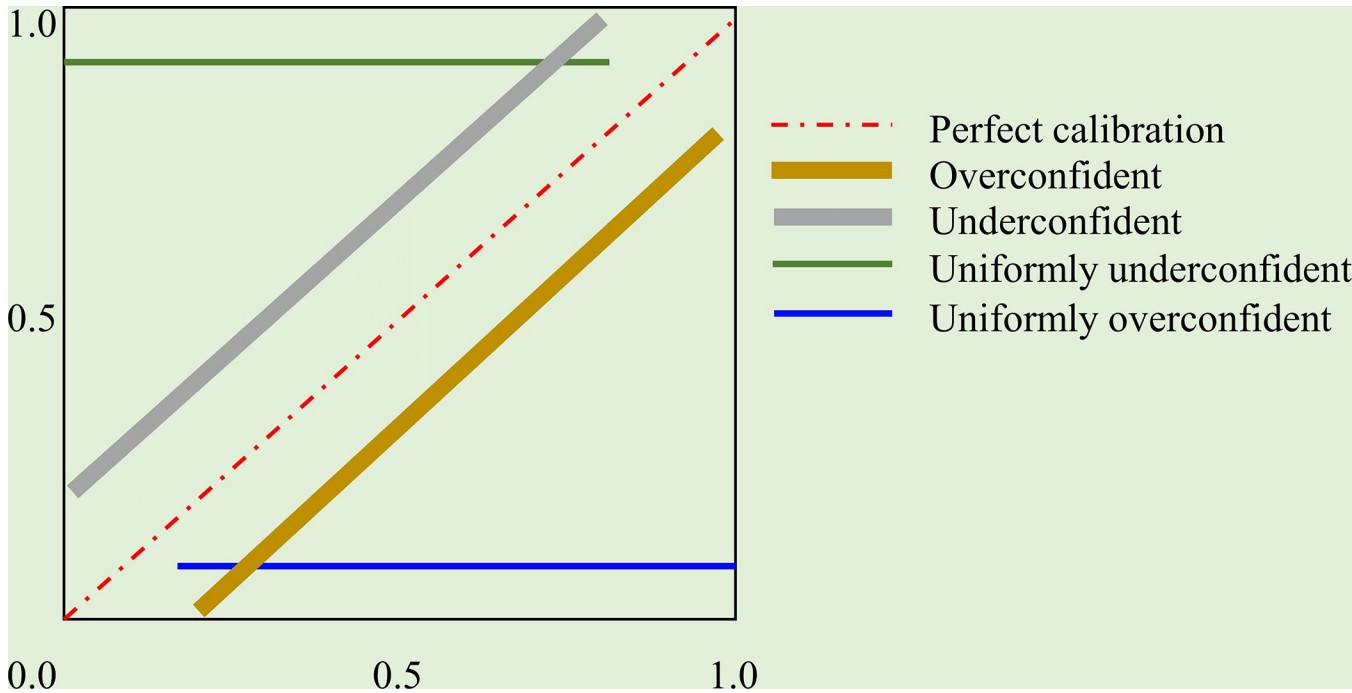

**Fig 2. A sample sketch of the reliability diagram shows perfectly calibrated, overconfident, underconfident, uniformly overconfident, and uniformly underconfident predictions.**

$$ECE = E_{p'}[abs((X' = X|P' = p) - p)] \tag{10}$$

In practice, the ECE metric is computed as the weighted average of the difference between the predicted probabilities and accuracy in each bin.

Here, $m$ is the total number of samples across all the probability bins. The value of ECE = 0 denotes the model is perfectly calibrated since *accuracy* ($C_z$) = *probability* ($C_z$) for all bins $z$.

## Calibration methods

The following calibration methods are used in this study: (i) Platt scaling, (ii) beta calibration, and (iii) spline calibration.

**Platt scaling.** Platt scaling [31] assumes a logistic relationship between the predicted probabilities ($z$) and true probability ($p$). It fits two parameters $\alpha$ and $\beta$ and is given by,

$$p = 1/(1 + \exp(-(\alpha + \beta z))) \tag{11}$$

The parameters $\alpha$ and $\beta$ are real-valued. The principal benefit of Platt scaling is that it needs very little data since it fits only two parameters. However, the limitation is there is a very restricted set of possible functions. That is, this method will deliver superior calibrated probabilities only if there exists a logistic relationship between $z$ and $p$.

**Beta calibration.** Literature studies reveal that Platt scaling-based calibration delivers suboptimal calibrated probabilities even compared to the original uncalibrated scores under circumstances when the classifiers produce heavily skewed score distributions. Under such circumstances, beta calibration [12] methods are shown to deliver superior calibration

performance as compared to Platt scaling. Beta calibration is given by,

$$p = \left(1 + \cfrac{1}{\exp(c)\frac{z^a}{(1-z)^b}}\right)^{-1} \tag{12}$$

The approach is similar to Platt scaling but with a couple of important improvements. It is a three-parameter family of curves (a, b, and c) compared to the 2-parameters used in Platt scaling. Beta calibration permits the diagonal y = x as one of the possible functions, so it would not affect an already calibrated classifier.

**Spline calibration.** Spline calibration [13] is proposed to be a robust, non-parametric calibration method that uses cubic smoothing splines to map the uncalibrated scores to true probabilities. Smoothing splines strike a balance between fitting the points well and having a smooth function. It uses a smoothed logistic function, so, the fit to the data is measured by likelihood and the smoothness refers to the integrated second derivative before the logistic transformation. A nuisance parameter trades-off smoothness for fit. It runs a lot of logistic regressions and picks the one with the best nuisance parameter. It transforms the data to provide appropriate scaling for over-confident models.

## Statistical analysis

Statistical analyses are performed to investigate if the performance differences between the models are statistically significant. We used a 95% confidence interval (CI) as the Wilson score interval for the MCC metric to compare the performance of the models trained and evaluated with datasets of varying imbalances. The CI values are also used to observe if there exists a statistically significant difference in the ECE metric before and after calibration. The Python StatsModels module is used to perform these evaluations.

## Results

### Classification performance achieved with Set-100 dataset

Recall that VGG-16, DenseNet-121, Inception-V3, and EfficientNet-B0 models are instantiated with their ImageNet-pretrained weights, truncated at their deepest convolutional layers, appended with the classification layers, and retrained on the Set-100 dataset constructed individually from (i)APTOS'19 fundus and (ii) Shenzhen TB CXR datasets, to classify them to their respective categories. This approach is followed to select the best-performing model that would subsequently be used to be retrained on the class-imbalance simulated (Set-20 and Set-60) datasets constructed from each of these data collections. The models are trained using a stochastic gradient descent optimizer with an initial learning rate of 1e-4 and momentum of 0.9. The learning rate is reduced whenever the validation loss plateaued. The best-performing model that delivered the least validation loss is used for class predictions. Table 2 summarizes the performance achieved by these models in this regard. S1 Fig shows the confusion matrix and AUPRC curves obtained using the DenseNet-121 and VGG-16 models, respectively, and S2 Fig shows the polar coordinates plot that summarizes the models' performance.

It is evident from the polar coordinates plot shown in S2 Fig that the models, in common, demonstrated higher values for AURPC and smaller values for the MCC for the reason how these measures are defined. The observation holds for both APTOS'19 fundus and Shenzhen TB CXR datasets. It is observed from Table 2 that, when retrained on the Set-100 dataset constructed from the APTOS'19 fundus dataset, the DenseNet-121 model demonstrated superior performance in terms of accuracy, F-score, and MCC metrics. The 95% CI for the MCC metric achieved by the DenseNet-121 model demonstrated a tighter error margin, hence, better

**Table 2. Test performance achieved by the models that are retrained on the Set-100 dataset, individually from the APTOS'19 fundus (n = 600) and Shenzhen TB CXR (n = 200) data collections.**

| Metric | Model | APTOS'19 fundus | Shenzhen TB CXR |
|--------|-------|-----------------|-----------------|
| Accuracy | VGG-16 | 0.7983 | **0.7850** |
| | D-121 | **0.8367** | 0.7000 |
| | I-V3 | 0.8033 | 0.5700 |
| | E-B0 | 0.8102 | 0.5920 |
| AUPRC | VGG-16 | **0.9723** | **0.8869** |
| | D-121 | 0.9290 | 0.8000 |
| | I-V3 | 0.9118 | 0.6215 |
| | E-B0 | 0.9216 | 0.6413 |
| F-score | VGG-16 | 0.8269 | **0.8054** |
| | D-121 | **0.8372** | 0.6202 |
| | I-V3 | 0.8097 | 0.4416 |
| | E-B0 | 0.8137 | 0.4734 |
| MCC | VGG-16 | 0.6321 | **0.5830**[*] |
| | | (0.5935, 0.6707) | (0.5146, 0.6514) |
| | D-121 | **0.6733**[*] | 0.4408 |
| | | (0.6357, 0.7109) | (0.3719, 0.5097) |
| | I-V3 | 0.6080 | 0.1577 |
| | | (0.5689, 0.6471) | (0.1071, 0.2083) |
| | E-B0 | 0.6258 | 0.1896 |
| | | (0.5870, 0.6646) | (0.1352, 0.2440) |

The value $n$ denotes the number of test samples. D-121, I-V3, and E-B0 represent the DenseNet-121, Inception-V3, and EfficientNet-B0 models, respectively. Data in parenthesis are 95% CI as the Wilson score interval provided for the MCC metric. The best performances are denoted by bold numerical values for each metric. The [*] denotes statistical significance ($p < 0.05$) compared to other models.

precision, and is observed to be significantly superior ($p < 0.05$) compared to that achieved with the VGG-16, Inception-V3, and EfficientNet-B0 models. Since the MCC metric provides a balanced measure of precision and recall, the DenseNet-121 model is selected as it demonstrated the best MCC metric, to be retrained and evaluated on the class-imbalance simulated (Set-20 and Set-60) datasets constructed from the APTOS'19 fundus dataset.

Considering the Shenzhen TB CXR dataset, the VGG-16 model demonstrated superior performance for accuracy, AUPRC, F-score, and a significantly superior value for the MCC metric ($p < 0.05$) compared to other models. Hence, the VGG-16 model is selected to be retrained and evaluated on the class-imbalance simulated datasets constructed from the Shenzhen TB CXR dataset.

## Calibration and classification performance measurements

Next, the best-performing DenseNet-121 and VGG-16 models are instantiated with their ImageNet-pretrained weights and retrained on the class-imbalance simulated (Set-20 and Set-60) datasets constructed from the APTOS'19 fundus and Shenzhen TB CXR datasets, respectively. The models are trained using a stochastic gradient descent optimizer with an initial learning rate of 1e-4 and momentum of 0.9. The learning rate is reduced whenever the validation loss plateaued. The best-performing model that delivered the least validation loss is used for prediction. Table 3 and Fig 3 show the ECE metric achieved using various calibration methods.

**Table 3. ECE metric achieved by the DenseNet-121 and VGG-16 models that are respectively retrained on the Set-20 and Set-60 datasets, individually from APTOS'19 fundus (n = 600) and Shenzhen TB CXR (n = 200) data collections.**

| Metric | Calibration method | APTOS'19 fundus | | | Shenzhen TB CXR | | |
|---|---|---|---|---|---|---|---|
| | | Set-20 | Set-60 | Set-100 | Set-20 | Set-60 | Set-100 |
| ECE | Platt | **0.0327* (0.0184, 0.047)** | **0.0409* (0.025, 0.0568)** | 0.0473 (0.0303, 0.0643) | 0.0832 (0.0449, 0.1215) | 0.0645 (0.0304, 0.0986) | **0.0463* (0.0171, 0.0755)** |
| | Beta | 0.0363 (0.0213, 0.0513) | 0.0435 (0.0271, 0.0599) | 0.0332 (0.0188, 0.0476) | 0.1021 (0.0601, 0.1441) | **0.0451* (0.0163, 0.0739)** | 0.0672 (0.0325, 0.1019) |
| | Spline | 0.0454 (0.0287, 0.0621) | 0.0439 (0.0275, 0.0603) | **0.0284* (0.0151, 0.0417)** | **0.0787* (0.0413, 0.1161)** | 0.0518 (0.021, 0.0826) | 0.0552 (0.0235, 0.0869) |
| | Baseline | 0.2124 (0.1796, 0.2452) | 0.1063 (0.0247, 0.0816) | 0.0518 (0.034, 0.0696) | 0.3237 (0.2588, 0.3886) | 0.0977 (0.0565, 0.1389) | 0.1378 (0.0900, 0.1856) |

The value *n* denotes the number of test samples. Baseline denotes uncalibrated probabilities. Data in parenthesis are 95% CI as the Wilson score interval provided for the ECE metric. The best performances are denoted by bold numerical values in the corresponding columns. The * denotes statistical significance ($p < 0.05$) compared to baseline.

From Table 3, we observe that no single calibration method delivered superior performance across all the datasets. For the Set-20 and Set-60 datasets constructed from the APTOS'19 fundus dataset, Platt calibration demonstrated the least ECE metric compared to other calibration methods. For the Set-100 dataset, spline calibration demonstrated the least ECE metric. The 95% CIs for the ECE metric achieved using the Set-20, Set-60, and Set-100 datasets demonstrated a tighter error margin and are observed to be significantly smaller ($p < 0.05$) compared to those obtained with uncalibrated, baseline probabilities.

A similar performance is observed with the Shenzhen TB CXR dataset. We observed that the spline, beta, and Platt calibration methods demonstrated the least ECE metric respectively for the Set-20, Set-60, and Set-100 datasets. The difference in the ECE metric is not statistically significant ($p > 0.05$) across the calibration methods. However, the 95% CIs for the ECE metric achieved using the Set-20, Set-60, and Set-100 datasets are observed to be significantly smaller ($p < 0.05$) compared to the uncalibrated, baseline model. This observation is evident from the polar coordinates plot shown in Fig 3 where the ECE values obtained with calibrated probabilities are smaller compared to those obtained with uncalibrated probabilities. The observation holds for the class-imbalance simulated datasets constructed from both APTOS'19 fundus and Shenzhen TB CXR datasets.

Fig 4 shows the reliability diagrams obtained using the uncalibrated and calibrated probabilities obtained using the Set-20 dataset constructed from (i) APTOS'19 fundus and (ii) Shenzhen TB CXR datasets. As observed from Fig 4A, the uncalibrated, baseline model is underconfident about its predictions since all the points are observed to lie above the diagonal line. Similar miscalibration issues are observed in Fig 4B for the Set-20 dataset constructed from the Shenzhen TB CXR dataset. As observed from the reliability diagram, the average probabilities of the fraction of disease-positive samples in the Shenzhen TB CXR Set-20 dataset are concentrated in the range [0.5 0.21]. This infers that all abnormal samples are misclassified as normal samples. However, the calibration methods attempted to rescale these uncalibrated probabilities to match their true occurrence likelihood and bring the points closer to the 45-degree line. The reliability diagrams for the other class-imbalance simulated datasets are given in S3 Fig.

Fig 5 and Table 4 summarize the performance achieved at the default classification threshold of 0.5 using the calibrated and uncalibrated probabilities for the Set-20, Set-60, and Set-100 datasets, constructed from the APTOS'19 fundus and Shenzhen TB CXR datasets. The calibration is performed using the best-performing calibration methods reported in Table 3.

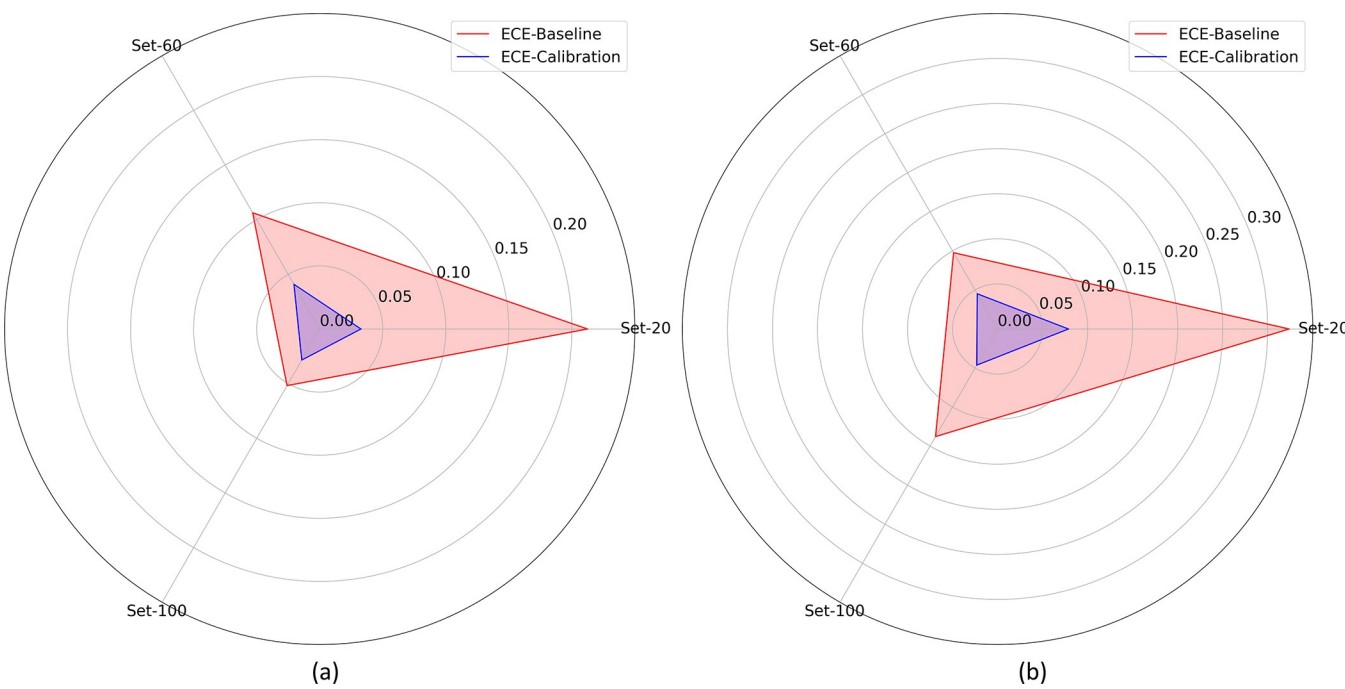

**Fig 3.** Polar coordinates plot showing the ECE metric achieved by the DenseNet-121 and VGG-16 models retrained on the Set-20, Set-60, and Set-100 datasets from (a) APTOS'19 fundus and (b) Shenzhen TB CXR datasets.

It is evident from the polar coordinates plot shown in Fig 5 that the MCC metric achieved using the calibrated probabilities for the Set-20, Set-60, and Set-100 datasets are higher compared to those achieved with the uncalibrated probabilities. This observation holds for both APTOS'19 fundus and Shenzhen TB CXR datasets. It is observed from Table 4 that, for the APTOS'19 fundus dataset, the MCC metric achieved using the calibrated probabilities for the Set-20 dataset is significantly superior ($p < 0.05$) compared to that achieved with the uncalibrated probabilities.

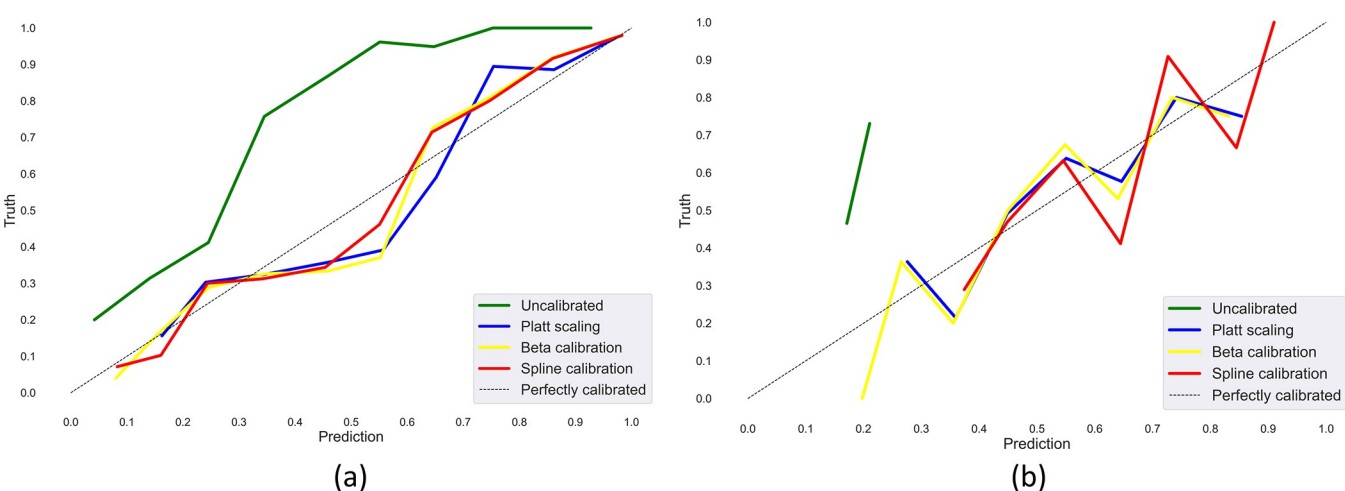

**Fig 4.** Reliability diagrams obtained using the uncalibrated and calibrated probabilities for the Set-20 dataset constructed from (a) APTOS'19 fundus and (b) Shenzhen TB CXR datasets.

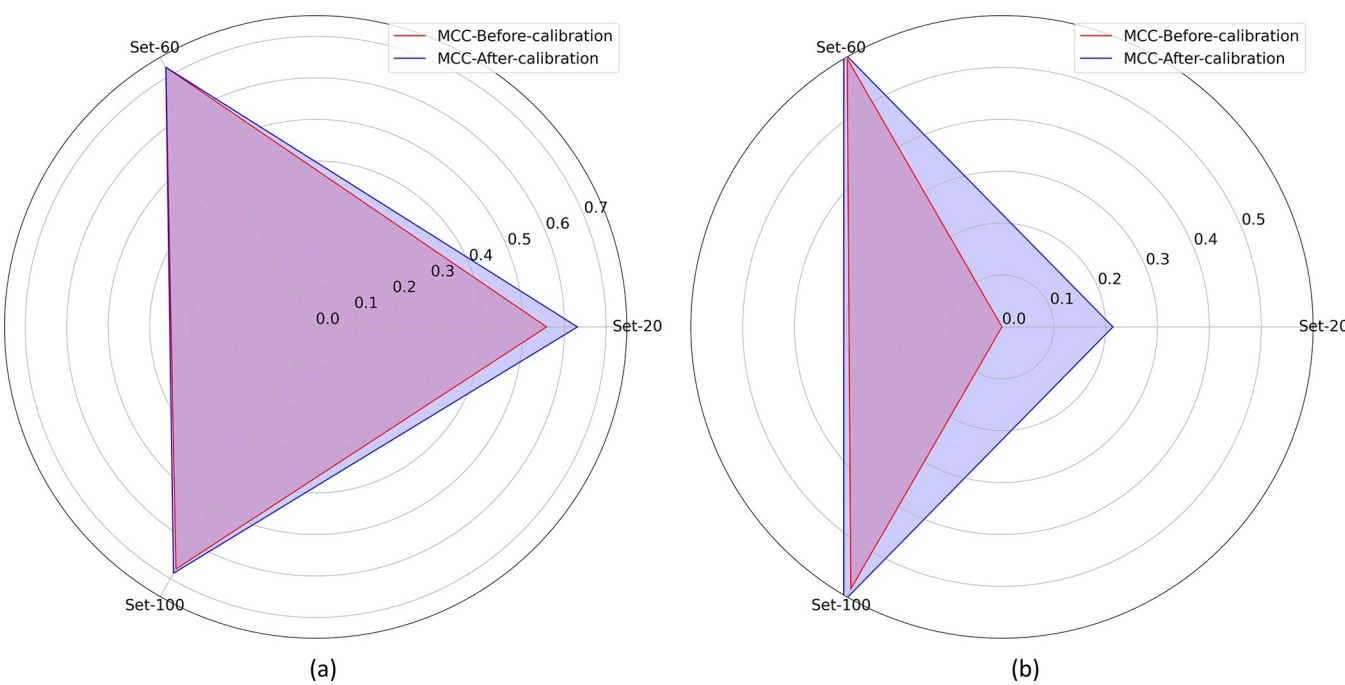

**Fig 5.** Polar coordinates plot showing the MCC metric achieved at the default operating threshold of 0.5, by the DenseNet-121 and VGG-16 models using calibrated and uncalibrated probabilities generated from Set-20, Set-60, and Set-100 datasets for (a) APTOS'19 fundus and (b) Shenzhen TB CXR data collections, respectively.

A similar performance is observed with the Set-20 and Set-60 datasets constructed from the Shenzhen TB CXR dataset. In particular, the F-score and MCC metric achieved with the uncalibrated probabilities is observed to be undefined. This is because the true positives (TPs) are 0 since all disease-positive samples are misclassified as normal samples. However, MCC values achieved with the calibrated probabilities are significantly higher ($p < 0.05$) compared to those achieved with the uncalibrated probabilities. This underscores the fact that calibration helped to significantly improve classification performance at the default classification threshold of 0.5.

**Table 4. Performance metrics achieved at the default operating threshold of 0.5, by the DenseNet-121 and VGG-16 models using calibrated (obtained using the best-performing calibration method from Table 3) and uncalibrated probabilities that are generated for Set-20, Set-60, and Set-100 datasets, constructed from the APTOS'19 fundus (n = 600) and Shenzhen TB CXR (n = 200) datasets, respectively.**

| Metric | APTOS'19 fundus | | | Shenzhen TB CXR | | |
|---|---|---|---|---|---|---|
| | Set-20 | Set-60 | Set-100 | Set-20 | Set-60 | Set-100 |
| Accuracy | **0.8117** | **0.8600** | **0.8417** | **0.6050** | **0.8050** | **0.8050** |
| | (0.7417) | (0.8500) | (0.8367) | (0.5000) | (0.7950) | (0.785) |
| AUPRC | 0.9034 | 0.9455 | 0.9290 | 0.6494 | **0.9004** | 0.8869 |
| | (0.9034) | (0.9455) | (0.9290) | (0.6494) | (0.9004) | (0.8869) |
| F-score | **0.7957** | **0.8789** | **0.8372** | **0.5635** | **0.804** | **0.8079** |
| | (0.6563) | (0.8289) | (0.8359) | (NA) | (0.8093) | (0.8054) |
| MCC | **0.6311**[*] | **0.7223** | **0.6850** | **0.2139**[*] | **0.6100** | **0.6103** |
| | (0.5569) | (0.7219) | (0.6733) | (NA) | (0.5968) | (0.583) |

The value $n$ denotes the number of test samples. Data in parenthesis denote the performance achieved with uncalibrated probabilities and data outside the parenthesis denotes the performance achieved with calibrated probabilities. The best performances are denoted by bold numerical values. The [*] denotes statistical significance ($p < 0.05$) compared to the performance obtained with uncalibrated probabilities.

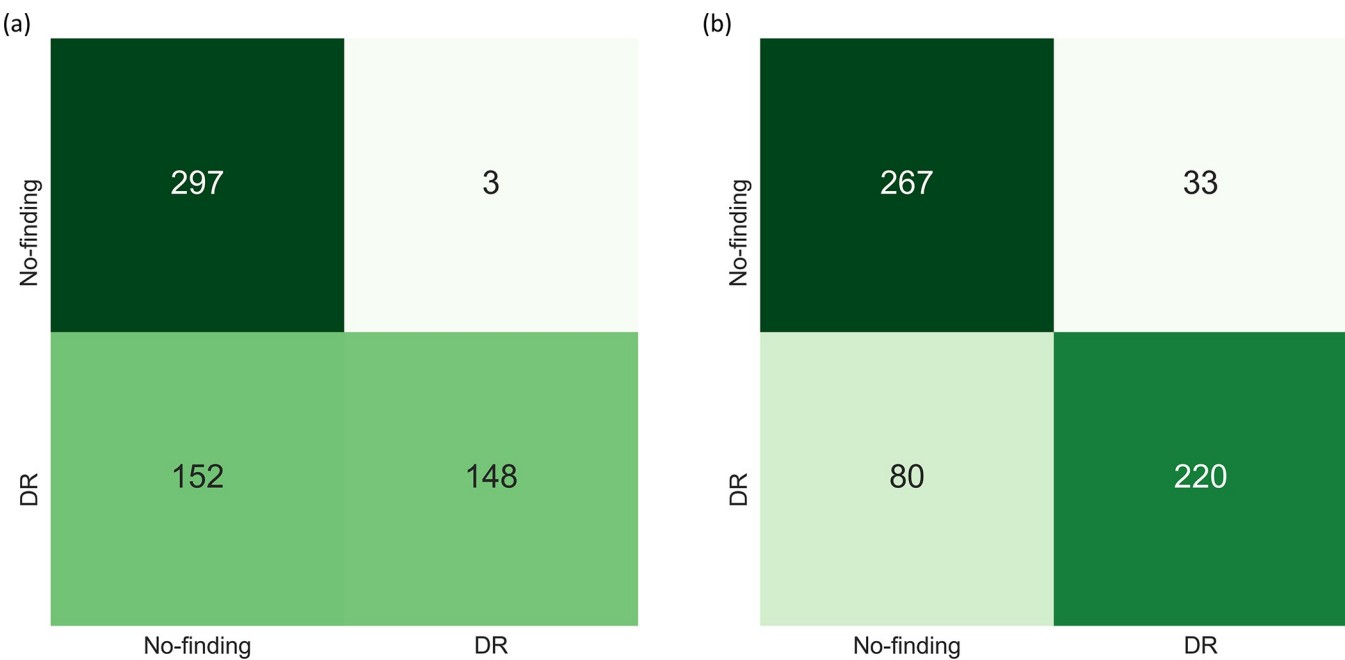

**Fig 6. Confusion matrices obtained using the uncalibrated and calibrated probabilities (from left to right) at the baseline threshold of 0.5 for the Set-20 dataset constructed from the APTOS'19 fundus dataset.**

Figs 6 and 7 show the confusion matrices obtained using the uncalibrated and calibrated probabilities, at the default classification threshold of 0.5, for the Set-20 dataset, individually constructed from the APTOS'19 fundus and Shenzhen TB CXR datasets. S4 and S5 Figs show the confusion matrices obtained for other class-imbalance simulated datasets.

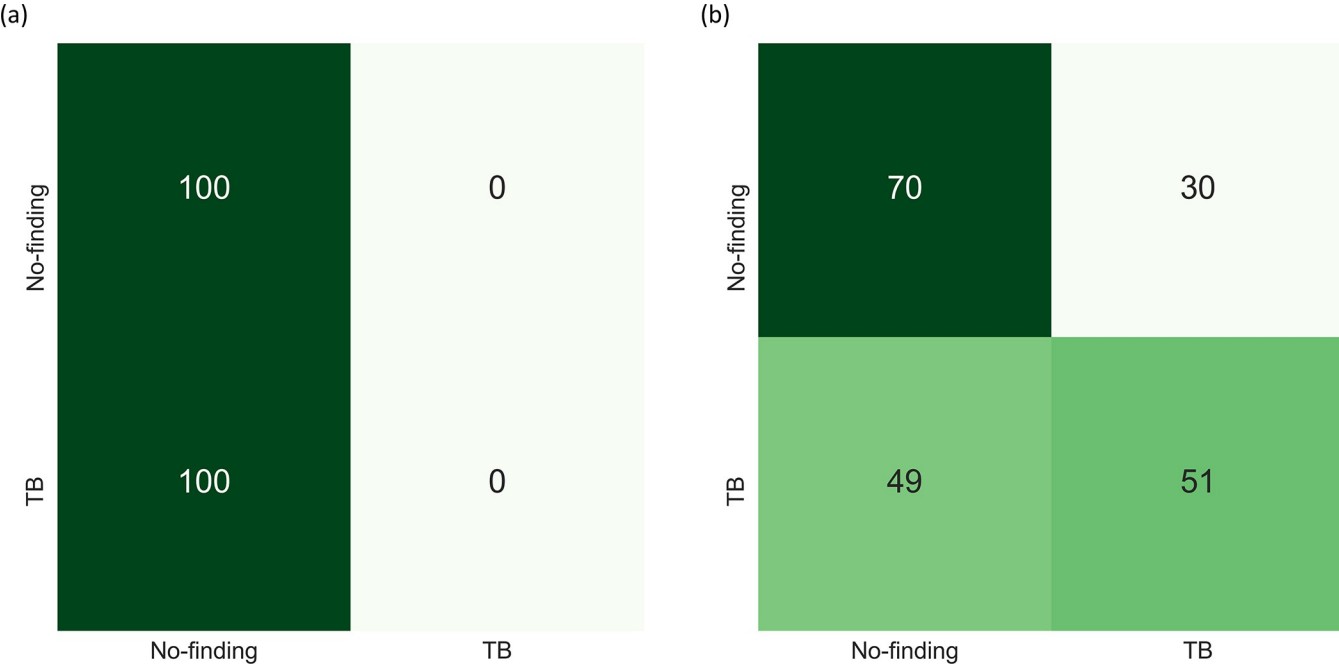

**Fig 7. Confusion matrices obtained with the uncalibrated and calibrated probabilities (from left to right) at the baseline threshold of 0.5 for the Set-20 dataset constructed from the Shenzhen TB CXR dataset.**

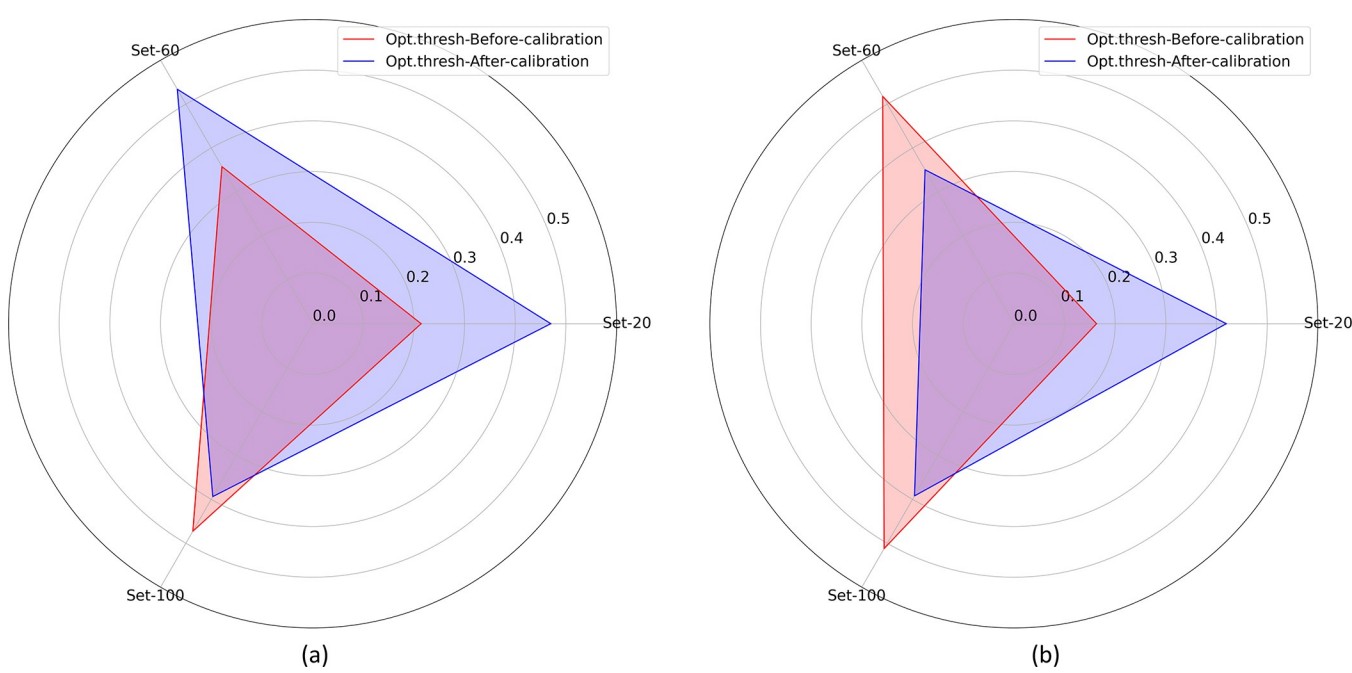

**Fig 8.** Polar coordinates plot showing the optimal threshold values identified from the PR curves using uncalibrated and calibrated probabilities generated from Set-20, Set-60, and Set-100 datasets for (a) APTOS'19 fundus and (b) Shenzhen TB CXR data collections.

Fig 8 and Table 5 summarize the optimal threshold values identified from the PR curves using the uncalibrated and calibrated probabilities. The probabilities are calibrated using the best-performing calibration method as reported in Table 3.

The polar coordinates plot shown in Fig 8 illustrates a difference in the optimal threshold values obtained before and after calibration. It is observed from Table 5 that the optimal threshold values are significantly different ($p < 0.05$) for the uncalibrated and calibrated probabilities obtained across the class-imbalance simulated datasets. The observation holds for both APTOS'19 fundus and Shenzhen TB CXR data collections. Fig 9 shows the PR curves with their optimal thresholds, obtained using the uncalibrated and calibrated probabilities for the Set-20 dataset, constructed from the APTOS'19 fundus and Shenzhen TB CXR datasets.

The PR curves for other class-imbalance simulated datasets are shown in S6 Fig. The performance obtained at these optimal threshold values is summarized in Table 6 and S7 Fig. It is evident from the polar coordinates plot shown in S7 Fig that, at the optimal threshold values derived from the PR curves, there is no significant difference in the MCC values obtained

**Table 5. Optimal threshold values identified from the PR curves using uncalibrated and calibrated probabilities (using the best-performing calibration method for the respective datasets).**

| Data | APTOS'19 fundus | | Shenzhen TB CXR | |
|---|---|---|---|---|
| | Opt. threshold (Uncalibrated) | Opt. threshold (Calibrated) | Opt. threshold (Uncalibrated) | Opt. threshold (Calibrated) |
| Set-20 | 0.2143 | 0.4701* (*Platt*) | 0.1632 | 0.4192* (*Spline*) |
| Set-60 | 0.3577 | 0.5339* (*Platt*) | 0.5177 | 0.3505* (*Beta*) |
| Set-100 | 0.4726 | 0.3937* (*Spline*) | 0.5121 | 0.3921* (*Platt*) |

The text in parentheses shows the best-performing calibration method used to produce calibrated probabilities. The * denotes statistical significance ($p < 0.05$) compared to the optimal threshold obtained with uncalibrated models.

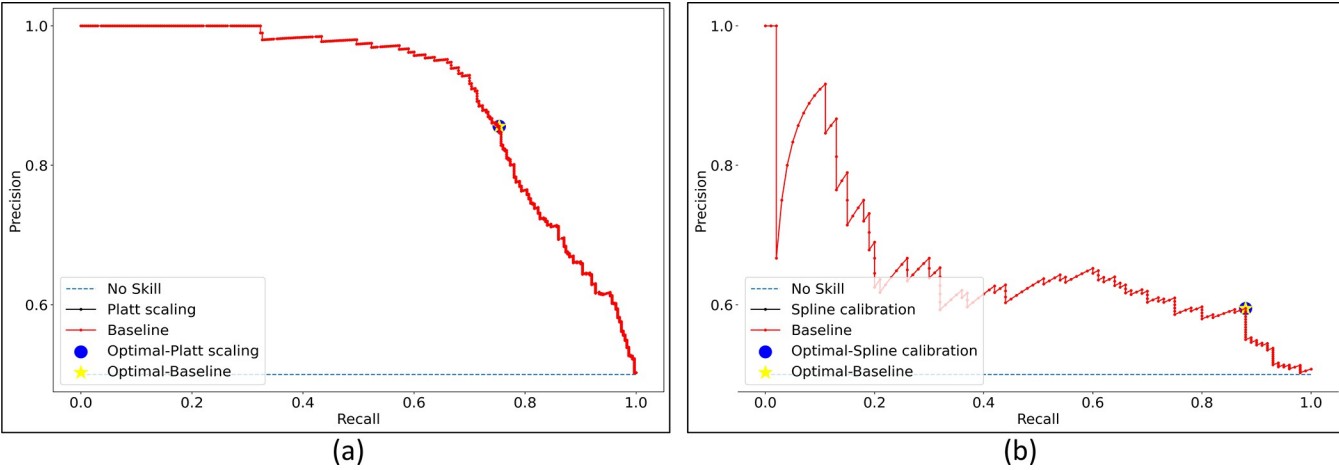

**Fig 9.** PR curves with their optimal thresholds obtained using the uncalibrated and calibrated probabilities for the Set-20 dataset, individually constructed from the (a) APTOS'19 fundus and (b) Shenzhen TB CXR datasets.

before and after calibration. This is also evident from Table 6 where, at the PR-guided optimal threshold, the classification performance obtained with the calibrated probabilities is not significantly superior ($p > 0.05$) compared to that obtained with the uncalibrated probabilities. This observation holds across the class-imbalance simulated datasets constructed from the APTOS'19 fundus and Shenzhen TB CXR collections. Figs 10 and 11 show the confusion matrices obtained using the uncalibrated and calibrated probabilities, at the optimal thresholds derived from the PR curves, for the Set-20 dataset, individually constructed from the APTOS'19 fundus and Shenzhen TB CXR collections. S8 and S9 Figs show the confusion matrices obtained for other class-imbalance simulated datasets.

We observed similar performances while repeating the aforementioned experiments with Set-40 (number of disease-positive samples is 40% of that in the normal class) and Set-80 (number of disease-positive samples is 80% of that in the normal class) datasets, individually constructed from the APTOS'19 fundus and Shenzhen TB CXR data collections. S1 Table shows the ECE metric achieved using various calibration methods for the Set-40 and Set-80

**Table 6. Performance metrics achieved at the optimal threshold values (from Table 6), by the DenseNet-121 and VGG-16 models using calibrated (using the best performing calibration method from Table 3) and uncalibrated probabilities generated for Set-20, Set-60, and Set-100 datasets, constructed from the APTOS'19 fundus (n = 600) and Shenzhen TB CXR (n = 200) datasets, respectively.**

| Metric | APTOS'19 fundus | | | Shenzhen TB CXR | | |
|---|---|---|---|---|---|---|
| | Set-20 | Set-60 | Set-100 | Set-20 | Set-60 | Set-100 |
| Accuracy | 0.8133 | 0.8683 | 0.8400 | **0.6400** | **0.8200** | 0.7950 |
| | (0.8133) | (0.8683) | (0.8400) | (0.6350) | (0.8150) | (0.7950) |
| AUPRC | 0.9034 | 0.9455 | 0.9290 | 0.6494 | 0.9091 | 0.8869 |
| | (0.9034) | (0.9455) | (0.9290) | (0.6494) | (0.9091) | (0.8869) |
| F-score | 0.8014 | 0.8612 | 0.8342 | **0.7097** | **0.8286** | 0.8110 |
| | (0.8014) | (0.8612) | (0.8342) | (0.7044) | (0.8230) | (0.8110) |
| MCC | 0.6312 | 0.7406 | 0.6802 | **0.3192** | **0.6432** | 0.5987 |
| | (0.6312) | (0.7406) | (0.6802) | (0.3059) | (0.6326) | (0.5987) |

Data in parenthesis denote the performance achieved with uncalibrated probabilities and data outside the parenthesis denotes the performance achieved with calibrated probabilities. The best performances are denoted by bold numerical values.

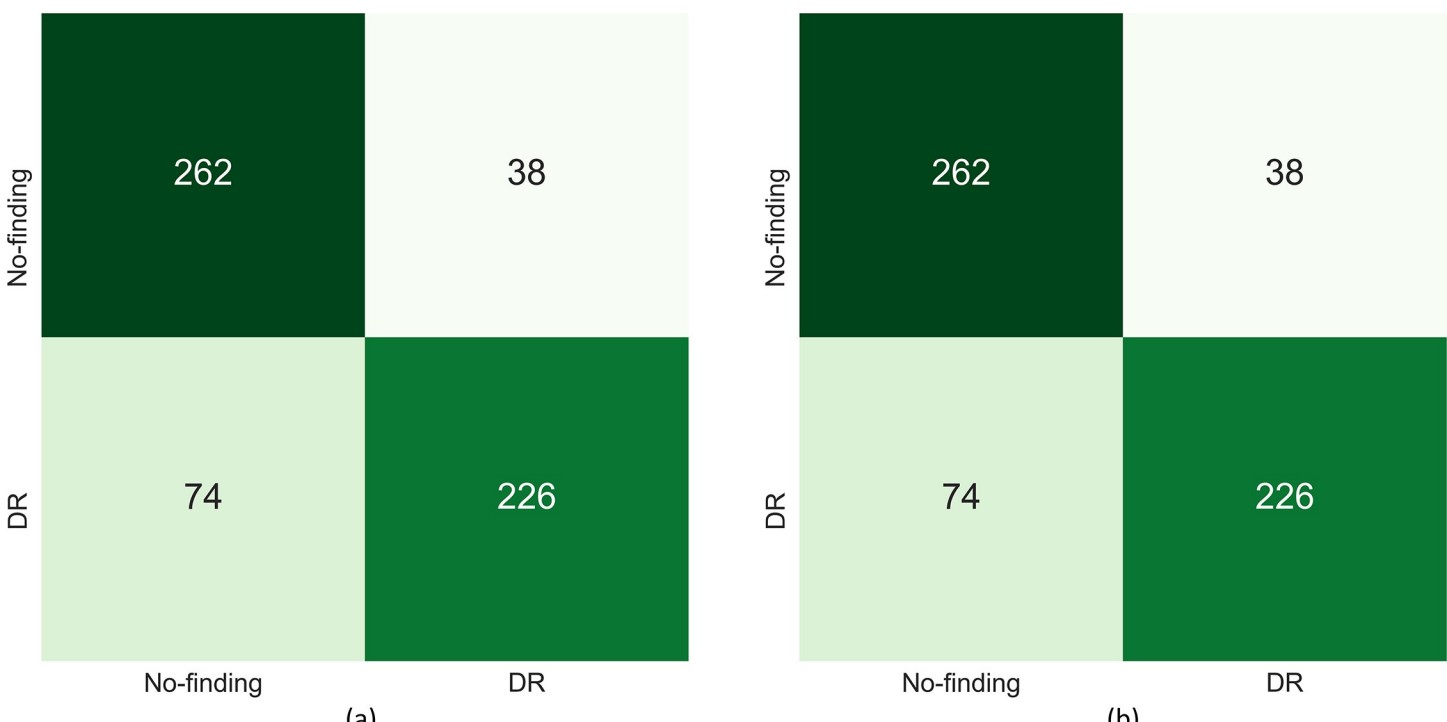

**Fig 10. Confusion matrices obtained using the uncalibrated and calibrated probabilities (from left to right) at the optimal thresholds derived from the PR curves (refer to Table 6) using the Set-20 dataset constructed from the APTOS'19 fundus dataset.**

datasets constructed from the APTOS'19 fundus and Shenzhen TB CXR data collections. S2 Table shows the performance achieved at the baseline operating threshold of 0.5 using the calibrated and uncalibrated probabilities for the Set-40 and Set-80 datasets. The calibration is performed using the best-performing calibration method as reported in the S1 Table. S3 Table shows the optimal threshold values identified from the PR curves using the uncalibrated and calibrated probabilities for the Set-40 and Set-80 datasets. S4 Table shows the performance obtained at the optimal threshold values identified from the PR curves for Set-40 and Set-80 datasets.

## Discussion and conclusions

We critically analyze and interpret the findings of our study as given below:

### Model selection

The method of selecting the most appropriate model from a collection of candidate models depends on the data size, type, characteristics, and behavior. It is worth noting that the DL models are pretrained on a large-scale collection of natural photographic images whose visual characteristics are distinct from medical images [16]. These models differ in several characteristics such as architecture, parameters, and learning strategies. Hence, they learn different feature representations from the data. For medical image classification tasks with sparse data availability, deeper models may not be always optimal since they may overfit the training data and demonstrate poor generalization [2]. It is therefore indispensable that for any given medical data, the most appropriate model should be identified that could help extract meaningful

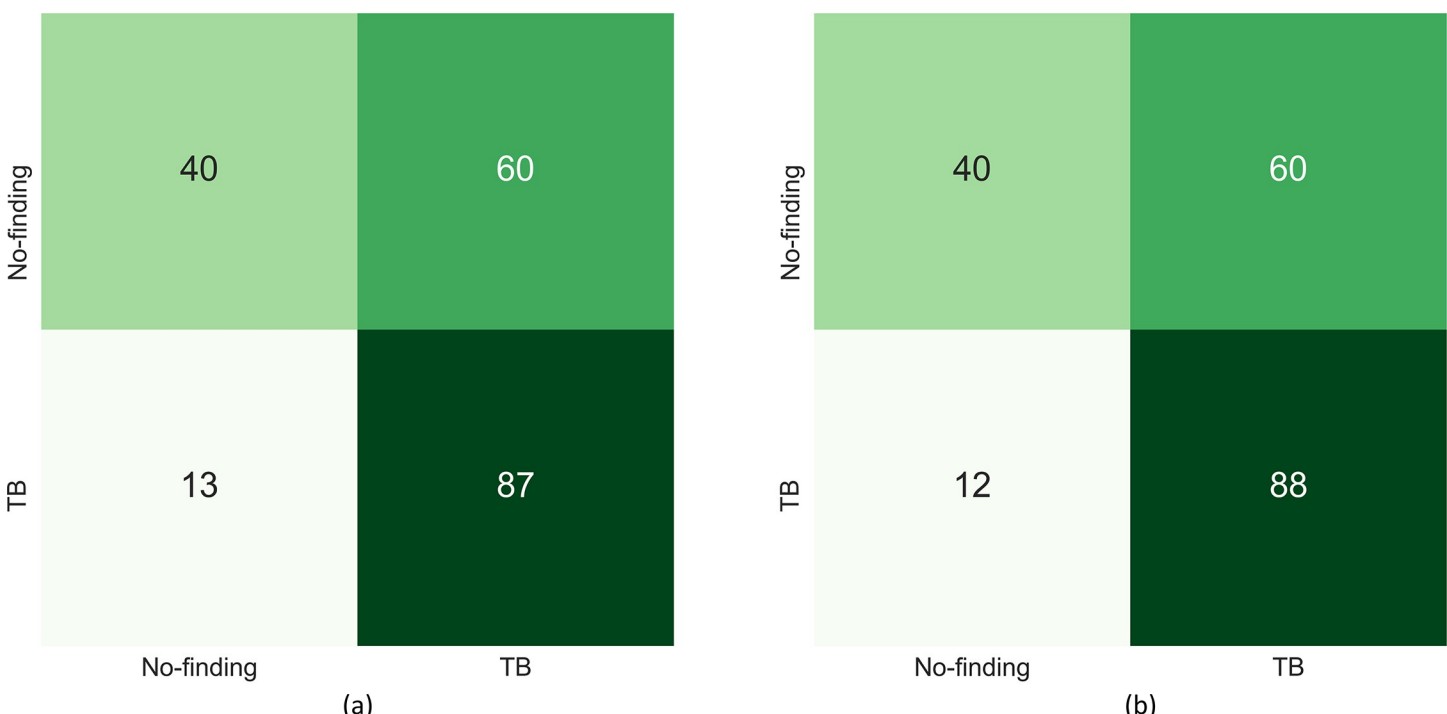

**Fig 11. Confusion matrices obtained with the uncalibrated and calibrated probabilities (from left to right) at their optimal thresholds derived from the PR curves (refer to Table 6) using the Set-20 dataset constructed from the Shenzhen TB CXR dataset.**

feature representations and deliver superior classification performance. In this study, we experimented with several DL models that delivered SOTA performance on medical image classification tasks and selected the best model that delivered superior performance. While using the best model for a given dataset, we observed that the performance with the test set improved with an increase in class balance. This observation holds for both APTOS'19 fundus and Shenzhen TB CXR datasets. The model demonstrated superior recall values with an increasing number of positive abnormal samples in the training set. This shows the model learned meaningful feature representations from the additional training samples in the positive abnormal class to correctly classify more abnormals in the test set.

## Simulating data imbalance

A review of the literature shows several studies that analyze the effect of calibration in a model trained with fixed-size data [9, 14, 15]. Until the time of writing this manuscript, to the best of our knowledge, we observed that no literature is available that explored the relationship between the calibration methods, degree of class imbalance, and model performance. Such an analysis would be significant, particularly considering medical image classification tasks, where there exist issues such as (i) low volume of disease samples and (ii) limited availability of expert annotations. In this study, we simulated class imbalance by dividing a balanced dataset into multiple datasets with varying degrees of imbalance of the positive disease samples. We observed that different calibration methods delivered improved calibration performance with different datasets. This underscores the fact that the performance obtained with a given calibration method depends on the (i) existing relationship between the predicted probabilities and the fraction of positive disease samples and (ii) if that calibration method would help map these uncalibrated probabilities to the true likelihood of occurrence of these samples.

### The values of AUPRC before and after calibration

We observed that irrespective of the calibration method, the value of AUPRC didn't change before and after calibration. This is because AUPRC provides a measure of discrimination [30]. This is a rank measure that helps to analyze if the observations are put in the best possible order. However, such an analysis does not ensure that the predicted probabilities would represent the true occurrence likelihood of events. On the other hand, calibration applies a transformation to map the uncalibrated probabilities to their true occurrence likelihood while maintaining the rank order. Therefore, the AUPRC values remained unchanged after calibration.

### PR-guided threshold and model performance

Unlike ROC curves, PR curves focus on model performance for the positive disease class samples that are low volume, high-impact events in a classification task. Hence, they are more useful where the positive disease class is significant compared to the negative class and are more informative than the ROC curves, particularly in imbalanced classification tasks [30]. We aimed to (i) identify an optimal PR-guided threshold for varying degrees of data imbalances and (ii) investigate if the classification performance obtained with these optimal thresholds derived from calibrated probabilities would be significantly superior ($p < 0.05$) compared to those derived from uncalibrated probabilities. We observed that, at the default classification threshold of 0.5, the classification performance achieved with the calibrated probabilities is significantly superior ($p < 0.05$) compared to that obtained with the uncalibrated probabilities. This holds when experimenting with the class-imbalance simulated datasets constructed from both APTOS'19 fundus and Shenzhen TB CXR data collections. This observation underscores the fact that, at the default classification threshold of 0.5, calibration helped to significantly improve classification performance. However, literature studies reveal that adopting the theoretical threshold of 0.5 may adversely impact performance in class imbalanced classification tasks that is common with medical images where the abnormal samples are considered rare events [26, 27]. Hence, we derived the optimal threshold from the PR curves.

We observed that the performance achieved with the PR-guided threshold derived from calibrated probabilities is not significantly superior ($p > 0.05$) compared to that derived from uncalibrated probabilities. It is important to note that calibration does not necessarily improve performance. The purpose of calibration is to rescale the predicted probabilities to reflect the true likelihood of occurrence of the class samples. The lack of association between calibration and model performance has also been reported in the literature [33] that demonstrates that the performance may not significantly improve after calibration. Therefore, model calibration guarantees the most reliable performance from a classifier, not necessarily the best performance for a given problem. In other words, the desired best performance depends on other factors such as data size, diversity, DL model selection, training strategy, etc. This performance is made more reliable by model calibration.

### Limitations and future work

The limitations of this study are: (i) We evaluated the performance of VGG-16, DenseNet-121, Inception-V3, and EfficientNet-B0 models, before and after calibration, toward classifying the datasets discussed in this study. With several DL models with varying architectural diversity being reported in the literature in recent times, future studies could focus on using multiple DL models and perform ensemble learning to learn improved predictions compared to any individual constituent model. (ii) We used PR curves to find the optimal threshold, however, there are other alternatives including ROC curve-based methods and manual threshold tuning.

The effect of optimal thresholds obtained from these methods on classification performance is an open research avenue. (iii) We used Platt scaling, beta calibration, and spline calibration methods in this study. However, we didn't use other popular calibration methods such as isotonic regression since we had limited data and our pilot studies showed overfitting with the use of isotonic regression-based calibration. This observation is identical to the results reported in the literature [32, 33]. (iv) We explored calibration performance with individual calibration methods. With a lot of research happening in calibration, new calibration algorithms and an ensemble of calibration methods may lead to improved calibration performance. (v) Calibration is used as a post-processing tool in this study. Future research could focus on proposing custom loss functions that incorporate calibration into the training process thereby alleviating the need for explicit training toward calibration.

## Supporting information

**S1 Fig. Test performance achieved by the models using the Set-100 dataset.** (a) and (b) confusion matrix achieved by the DenseNet-121 and VGG-16 models, respectively, using the APTOS'19 fundus and Shenzhen TB CXR data collections; (c) and (d) AUPRC curves achieved by the DenseNet-121 and VGG-16 models, respectively, using the APTOS'19 fundus and Shenzhen TB CXR data collections.
(TIF)

**S2 Fig.** Polar coordinates plot showing the test performance achieved by the models retrained on the Set-100 dataset from (a) APTOS'19 fundus and (b) Shenzhen TB CXR datasets.
(TIF)

**S3 Fig. Reliability diagrams obtained using the uncalibrated and calibrated probabilities for the Set-40, Set-60, Set-80, and Set-100 datasets.** (a), (c), (e), and (g) shows the reliability diagrams obtained respectively using the. Set-40, Set-60, Set-80, and Set-100 datasets constructed from APTOS'19 fundus dataset; (b), (d), (f), and (h) show the reliability diagrams obtained respectively using the Set-40, Set-60, Set-80, and Set-100 datasets constructed from Shenzhen TB CXR dataset.
(TIF)

**S4 Fig.** Confusion matrices obtained using the uncalibrated and calibrated probabilities (from left to right) at the baseline threshold of 0.5 for the Set-40, Set-60, and Set-80 datasets constructed from the APTOS'19 fundus dataset. (a), (c), and (e) show the confusion matrices obtained using uncalibrated probabilities; (b), (d), and (f) show the confusion matrices obtained using calibrated probabilities.
(TIF)

**S5 Fig.** Confusion matrices obtained using the uncalibrated and calibrated probabilities (from left to right) at the baseline threshold of 0.5 for the Set-40, Set-60, and Set-80 datasets constructed from the Shenzhen TB CXR dataset. (a), (c), and (e) show the confusion matrices obtained using uncalibrated probabilities; (b), (d), and (f) show the confusion matrices obtained using calibrated probabilities.
(TIF)

**S6 Fig. PR curves with their optimal thresholds obtained using the uncalibrated and calibrated probabilities for the Set-40, Set-60, Set-80, and Set-100 datasets.** (a), (c), (e), and (g) shows the PR curves obtained respectively using the Set-40, Set-60, Set-80, and Set-100 datasets from APTOS'19 fundus dataset; (b), (d), (f), and (h) show the PR curves obtained respectively

using the Set-40, Set-60, Set-80, and Set-100 datasets from Shenzhen TB CXR dataset.
(TIF)

**S7 Fig.** Polar coordinates plot showing the MCC metric achieved at the optimal operating thresholds, by the DenseNet-121 and VGG-16 models using calibrated and uncalibrated probabilities generated from Set-20, Set-60, and Set-100 datasets for (a) APTOS'19 fundus and (b) Shenzhen TB CXR data collections, respectively.
(TIF)

**S8 Fig.** Confusion matrices obtained using the uncalibrated and calibrated probabilities (from left to right) at the optimal thresholds derived from the PR curves for the Set-40, Set-60, and Set-80 datasets constructed from the APTOS'19 fundus dataset. (a), (c), and (e) show the confusion matrices obtained using uncalibrated probabilities; (b), (d), and (f) show the confusion matrices obtained using calibrated probabilities.
(TIF)

**S9 Fig Confusion matrices obtained using the uncalibrated and calibrated probabilities (from left to right) at the optimal thresholds derived from the PR curves for the Set-40, Set-60, and Set-80 datasets constructed from the Shenzhen TB CXR dataset (a), (c), and (e) show the confusion matrices obtained using uncalibrated probabilities; (b), (d), and (f) show the confusion matrices obtained using calibrated probabilities.**
(TIF)

**S1 Table. ECE metric achieved by the DenseNet-121 and VGG-16 models that are respectively retrained on the Set-40 and Set-80 datasets, individually from APTOS'19 fundus (n = 600) and Shenzhen TB CXR (n = 200) image collections.** The value $n$ denotes the number of test samples. Data in parenthesis are 95% CI as the Wilson score interval provided for the ECE metric. The best performances are denoted by bold numerical values in the corresponding columns.
(PDF)

**S2 Table. Performance metrics achieved at the baseline threshold of 0.5, by the DenseNet-121 and VGG-16 models using calibrated (using the best performing calibration method from Table 4) and uncalibrated probabilities generated for Set-40 and Set-80 datasets from the APTOS'19 fundus (n = 600) and Shenzhen TB CXR (n = 200) image collections, respectively.** Data in parenthesis denote the performance achieved with uncalibrated probabilities and data outside the parenthesis denotes the performance achieved with calibrated probabilities. The best performances are denoted by bold numerical values in the corresponding columns.
(PDF)

**S3 Table. Optimal threshold values identified from the PR curves using uncalibrated and calibrated probabilities (using the best-performing calibration method for the respective datasets) for Set-40 and Set-80 datasets.** The text in parentheses shows the best-performing calibration method used to produce calibrated probabilities.
(PDF)

**S4 Table. Performance metrics achieved at the optimal threshold values (from Table 3), by the DenseNet-121 and VGG-16 models using calibrated (using the best performing calibration method from Table 4) and uncalibrated probabilities generated for Set-40 and Set-80 datasets from the APTOS 2019 fundus (n = 600) and Shenzhen TB CXR (n = 200) datasets, respectively.** Data in parenthesis denote the performance achieved with uncalibrated

probabilities and data outside the parenthesis denotes the performance achieved with calibrated probabilities. The best performances are denoted by bold numerical values.
(PDF)

## Author Contributions

**Conceptualization:** Sivaramakrishnan Rajaraman, Prasanth Ganesan, Sameer Antani.

**Data curation:** Sivaramakrishnan Rajaraman.

**Formal analysis:** Sivaramakrishnan Rajaraman, Sameer Antani.

**Funding acquisition:** Sameer Antani.

**Investigation:** Sameer Antani.

**Methodology:** Sivaramakrishnan Rajaraman, Prasanth Ganesan.

**Project administration:** Sameer Antani.

**Resources:** Sameer Antani.

**Supervision:** Sameer Antani.

**Validation:** Sivaramakrishnan Rajaraman.

**Visualization:** Sivaramakrishnan Rajaraman.

**Writing – original draft:** Sivaramakrishnan Rajaraman.

**Writing – review & editing:** Sivaramakrishnan Rajaraman, Prasanth Ganesan, Sameer Antani.

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
