## [Decision Letter · Decision Letter 0]

16 Nov 2021

PONE-D-21-32609Deep learning model calibration for improving performance in class-imbalanced medical image classification tasksPLOS ONE

Dear Dr. Rajaraman,

Thank you for submitting your manuscript to PLOS ONE. After careful consideration, we feel that it has merit but does not fully meet PLOS ONE’s publication criteria as it currently stands. Therefore, we invite you to submit a revised version of the manuscript that addresses the points raised during the review process.

ACADEMIC EDITOR: Based on the comments from the reviewers and my own assessment I recommend major revisions for the article

We look forward to receiving your revised manuscript.

Kind regards,

Thippa Reddy Gadekallu

Academic Editor

PLOS ONE

Journal Requirements:

"This study is supported by the Intramural Research Program (IRP) of the National Library of Medicine (NLM) and the National Institutes of Health (NIH)."

"This study is supported by the Intramural Research Program (IRP) of the National

Library of Medicine (NLM) and the National Institutes of Health (NIH). The intramural research scientists (authors) at the NIH dictated study design, data collection, data analysis, decision to publish and preparation of the manuscript."

4. We note that Figure 1 in your submission contain copyrighted images. All PLOS content is published under the Creative Commons Attribution License (CC BY 4.0), which means that the manuscript, images, and Supporting Information files will be freely available online, and any third party is permitted to access, download, copy, distribute, and use these materials in any way, even commercially, with proper attribution. For more information, see our copyright guidelines: http://journals.plos.org/plosone/s/licenses-and-copyright.

Reviewers' comments:

Reviewer's Responses to Questions

**Comments to the Author**

1. Is the manuscript technically sound, and do the data support the conclusions?

Reviewer #1: Yes

Reviewer #2: Partly

2. Has the statistical analysis been performed appropriately and rigorously? 

Reviewer #1: No

Reviewer #2: Yes

3. Have the authors made all data underlying the findings in their manuscript fully available?

Reviewer #1: No

Reviewer #2: Yes

4. Is the manuscript presented in an intelligible fashion and written in standard English?

Reviewer #1: Yes

Reviewer #2: No

5. Review Comments to the Author

Reviewer #1: The proposed work presents a deep learning-based calibration to improve the performance of medical classification tasks. Thus, it is not a novel research area. In addition, the manuscript has several other concerns.

The described architecture is very abstract and needs a detailed explanation of the step-wise process carried out. I recommend authors to add a detailed workflow describing the proposed approach.

The literature review carried out for the proposed work is outdated and needs the referral of some of the latest research works published in the last three years.

I recommend authors to use the benchmark dataset and perform similar experiments and discuss the comparison. Authors can create a customized dataset and describe the data collection process in detail using a process flow diagram if a benchmark dataset is not available.

I recommend authors to add a layered architecture and detailed work to give more clarity to readers about the proposed system.

I recommend authors to add limitations in detail(instead of abstract information) of the proposed system and future directions.

The resolution of all figures is a concern. I recommend authors to redraw most of the images to match the journal standards.

Reviewer #2: - Reorganize the introduction, trying to explain every word of the title.

- Add separate section Literature Review.

- The quality of the figures can be improved more in the results section. Figures should be eye-catching. It will enhance the interest of the reader.

- What are the computational resources reported in the state of the art for the same purpose?

- Please cite each equation and clearly explain its terms.

- What are the evaluations used for the verification of results?

- Authors should consider most recent literature in the related work which is missing in this article.

- In this manuscript, author intentions are not clear and ambiguous, and English used throughout the manuscript must be checked under the guidance of an expert.

6. PLOS authors have the option to publish the peer review history of their article (what does this mean?). If published, this will include your full peer review and any attached files.

Reviewer #1: No

Reviewer #2: **Yes: **Dr. Kadiyala Ramana

---

## [Author Response · Author response to Decision Letter 0]

24 Nov 2021

Response to the Editor:

We render our sincere thanks to the Editor for arranging peer review and encouraging resubmission of our manuscript. To the best of our knowledge and belief, we have addressed the concerns of the Editor and the reviewers in the revised manuscript.

Q1: When submitting your revision, we need you to address these additional requirements. 1. Please ensure that your manuscript meets PLOS ONE's style requirements, including those for file naming. The PLOS ONE style templates can be found at https://journals.plos.org/plosone/s/file?id=wjVg/PLOSOne_formatting_sample_main_body.pdf, https://journals.plos.org/plosone/s/file?id=ba62/PLOSOne_formatting_sample_title_authors_affiliations.pdf. 

Author response: We have formatted the manuscript per the templates recommended by the Editor. 

Q2: We note that the grant information you provided in the ‘Funding Information’ and ‘Financial Disclosure’ sections do not match. When you resubmit, please ensure that you provide the correct grant numbers for the awards you received for your study in the ‘Funding Information’ section.

Author response: SA and SR’s research is supported by the Intramural Research Program (IRP) of the National Library of Medicine (NLM) and the National Institutes of Health (NIH). PG received no financial compensation for this work. We do not have specific grant numbers. 

Q3: Thank you for stating the following in the Acknowledgments Section of your manuscript: "This study is supported by the Intramural Research Program (IRP) of the National Library of Medicine (NLM) and the National Institutes of Health (NIH)." We note that you have provided funding information that is not currently declared in your Funding Statement. However, funding information should not appear in the Acknowledgments section or other areas of your manuscript. We will only publish funding information present in the Funding Statement section of the online submission form. Please remove any funding-related text from the manuscript and let us know how you would like to update your Funding Statement. Currently, your Funding Statement reads as follows: "This study is supported by the Intramural Research Program (IRP) of the National Library of Medicine (NLM) and the National Institutes of Health (NIH). The intramural research scientists (authors) at the NIH dictated study design, data collection, data analysis, decision to publish and preparation of the manuscript." Please include your amended statements within your cover letter; we will change the online submission form on your behalf.

Author response: We have removed the Acknowledgment section (and included text) per the Editor’s recommendation. We hereby agree to include the following modified statements under the “Funding Information and Financial Disclosure” sections in the online submission form.

 “This study is supported by the Intramural Research Program (IRP) of the National Library of Medicine (NLM) and the National Institutes of Health (NIH). PG received no financial compensation for this work. 

Q4: We note that Figure 1 in your submission contain copyrighted images. All PLOS content is published under the Creative Commons Attribution License (CC BY 4.0), which means that the manuscript, images, and Supporting Information files will be freely available online, and any third party is permitted to access, download, copy, distribute, and use these materials in any way, even commercially, with proper attribution. For more information, see our copyright guidelines: http://journals.plos.org/plosone/s/licenses-and-copyright. We require you to either (1) present written permission from the copyright holder to publish these figures specifically under the CC BY 4.0 license, or (2) remove the figures from your submission: a. You may seek permission from the original copyright holder of Figure 1 to publish the content specifically under the CC BY 4.0 license. We recommend that you contact the original copyright holder with the Content Permission Form (http://journals.plos.org/plosone/s/file?id=7c09/content-permission-form.pdf) and the following text: “I request permission for the open-access journal PLOS ONE to publish XXX under the Creative Commons Attribution License (CCAL) CC BY 4.0 (http://creativecommons.org/licenses/by/4.0/). Please be aware that this license allows unrestricted use and distribution, even commercially, by third parties. Please reply and provide explicit written permission to publish XXX under a CC BY license and complete the attached form.” Please upload the completed Content Permission Form or other proof of granted permissions as an "Other" file with your submission. In the figure caption of the copyrighted figure, please include the following text: “Reprinted from [ref] under a CC BY license, with permission from [name of publisher], original copyright [original copyright year].” b. If you are unable to obtain permission from the original copyright holder to publish these figures under the CC BY 4.0 license or if the copyright holder’s requirements are incompatible with the CC BY 4.0 license, please either i) remove the figure or ii) supply a replacement figure that complies with the CC BY 4.0 license. Please check copyright information on all replacement figures and update the figure caption with source information. If applicable, please specify in the figure caption text when a figure is similar but not identical to the original image and is therefore for illustrative purposes only. 

Author response: Fig. 1 in the previous version of the manuscript was merely a collage of selected image samples from the datasets used in this study. All datasets are publicly available. We have cited the sources in the body of the manuscript. However, to avoid any impression of copyright violation, we have removed this figure in the revised manuscript per the Editor’s suggestions. 

Response to Reviewer #1:

We thank the reviewer for the valuable comments on this study.

Q1: The proposed work presents a deep learning-based calibration to improve the performance of medical classification tasks. Thus, it is not a novel research area. In addition, the manuscript has several other concerns. The described architecture is very abstract and needs a detailed explanation of the step-wise process carried out. I recommend authors to add a detailed workflow describing the proposed approach.

Author response: We would like to clarify the rationale for the study. Class-imbalanced training is common in medical imagery where the number of abnormal samples is considerably small compared to the normal samples. In such class-imbalanced situations, reliable training of deep neural networks continues to be a major challenge because the model may be biased toward the majority normal class. We agree with the reviewer that model calibration is an established approach to alleviate some of these effects. At this time, to the best of our knowledge, we find that no literature is available that explored the relationship between the calibration methods, degree of class imbalance, and model performance. Neither there exists any literature that guides whether or when such calibration would be beneficial. This is the novel contribution of our work.

 To this end, we perform (i) systematic analysis of the effect of model calibration using various deep learning classifier backbones, and (ii) study the impact of calibration for varying degrees of imbalances in the dataset used for training, calibration methods, two classification thresholds, namely, default threshold of 0.5, and optimal threshold from precision-recall (PR) curves, respectively. The architectures that we used are established off-the-shelf models with custom fine-tuning. Appropriate references are provided in lines 68, 69, 93, and 94 in this regard to avoid repetitive text on these well-known models. 

Q2: The literature review carried out for the proposed work is outdated and needs the referral of some of the latest research works published in the last three years.

Author response: We wish to confirm that we performed an extensive review and cited the most important studies on the calibration of deep learning models. These works include those published in reputed journals in 2020 and 2021. The concept of calibrating deep learning models is itself a less often discussed topic and does not have adequate literature. This is one reason why we wanted to address it in our current manuscript. 

Q3: I recommend authors to use the benchmark dataset and perform similar experiments and discuss the comparison. Authors can create a customized dataset and describe the data collection process in detail using a process flow diagram if a benchmark dataset is not available.

Author response: Thanks for these comments. The publicly available datasets used in our study are selected for their size and those that have been widely used [2, 3, 6, 7, 22]. We believe that the results obtained with these are adequate to support our findings.

Q4: I recommend authors to add a layered architecture and detailed work to give more clarity to readers about the proposed system.

Author response: We wish to reiterate our response to Q1. The architectures that we used are established off-the-shelf models with custom fine-tuning. Adequate references are provided in lines 68, 69, 93, and 94 in this regard. We try to avoid repetitive text on such well-known models, so we provided appropriate citations. 

Q5: I recommend authors to add limitations in detail(instead of abstract information) of the proposed system and future directions.

Author response: Agreed. The limitations of the current study and the scope for future work are discussed under the “Limitations and future work” section (lines 521 – 536) in the revised manuscript. 

Q6: The resolution of all figures is a concern. I recommend authors to redraw most of the images to match the journal standards.

Author response: Thanks. Our current resolution is 600dpi which is much above the limit of standards recommended by PLOS ONE. However, we have not converted the images into vector format (SVG). All figures are checked and converted using the PACE tool recommended by PLOS ONE during submission. We hope this addresses the resolution issue of the reviewer.

Response to Reviewer #2:

We render our sincere thanks to the reviewer for the valuable comments and appreciation of our study. To the best of our knowledge and belief, we have addressed the reviewer’s concerns. 

Q1: Reorganize the introduction, trying to explain every word of the title.

Author response: Thanks for these suggestions. The impact of deep learning in computer vision is discussed in lines 38 – 45. The adverse effects of class-imbalanced training and the existing methods are discussed in lines 46 – 56. Details considering the need for model calibration, the existing literature and its limitations, the need to perform a comprehensive analysis of the relationship between the degree of data imbalance, the calibration methods, and the effect of the classification threshold on model performance pre- and post-calibration are discussed in lines 57 – 83. The contributions of this study are discussed in lines 84 – 101. 

Q2: Add separate section Literature Review.

Author response: Thanks for these comments. As mentioned in the PLOS ONE submission guidelines for authors, the introduction should include a brief review of the key literature. They do not necessitate a separate section for the literature review. In this regard, we performed an extensive review and included the most important studies on the calibration of deep learning models in the introduction. These works include those published in reputed journals in 2020 and 2021. The concept of calibrating deep learning models is itself a less often discussed topic and does not have ubiquitous literature. This is one reason why we wanted to address it in our current manuscript.

Q3: The quality of the figures can be improved more in the results section. Figures should be eye-catching. It will enhance the interest of the reader.

Author response: Thanks. We believe the current figures we have are all self-explanatory. Our current resolution is 600dpi which is much above the limit of PLOS ONE recommended standards. However, we have not converted the images into vector format (SVG). All figures are checked and converted using the PACE tool recommended by PLOS ONE during submission. We hope this addresses the resolution issue of the reviewer.

Q4: What are the computational resources reported in the state of the art for the same purpose? Please cite each equation and clearly explain its terms.

Author response: Thanks for these comments. PLOS ONE does not require mentioning the equation numbers within the text. We made sure to number each equation in the revised manuscript per submission guidelines. We ensured that the parameters mentioned in the equations are discussed within the text. The methods discussed in the literature used Python scripts to propose calibration methods and well-known deep learning models to study calibration effects. Adequate references are included for each calibration method and the deep learning models discussed in this study. 

Q5: What are the evaluations used for the verification of results?

Author response: The performance of the deep learning models, before and after calibration, is evaluated in terms of the expected calibration error, accuracy, area under the precision-recall curve, F-score, and Matthews correlation coefficient. These are discussed under the “evaluation metrics” (lines 159 – 166) and quantitative evaluation of calibration (lines 210 – 216) sections. 

Q6: Authors should consider most recent literature in the related work which is missing in this article.

Author response: Thanks for these comments. We wish to confirm that we performed an extensive review and cited the most important studies on the calibration of deep learning models. These works include those published in reputed journals in 2020 and 2021. The concept of calibrating deep learning models is itself a less often discussed topic and does not have exhaustive literature. This is one reason why we wanted to address it in our current manuscript. 

Q7: In this manuscript, author intentions are not clear and ambiguous, and English used throughout the manuscript must be checked under the guidance of an expert.

Author response: Thanks for these comments. We would like to clarify the rationale for the study. Class-imbalanced training is common in medical imagery where the number of abnormal samples is considerably small compared to the normal samples. In such class-imbalanced situations, reliable training of deep neural networks continues to be a major challenge because the model may be biased toward the majority normal class. Though model calibration is an established approach to alleviate some of these effects, there is insufficient analysis explaining whether and when calibrating a model would be beneficial. Until the time of writing this manuscript, to the best of our knowledge, we find that no literature is available that explored the relationship between the calibration methods, degree of class imbalance, and model performance. To this end, we perform (i) systematic analysis of the effect of model calibration using various deep learning classifier backbones, and (ii) studied several variations including the degree of imbalances in the dataset used for training, calibration methods, two classification thresholds, namely, default threshold of 0.5, and optimal threshold from precision-recall (PR) curves, respectively. Our results indicate that at the default classification threshold of 0.5, the performance achieved through calibration is significantly superior (p < 0.05) to using uncalibrated probabilities. However, at the PR-guided threshold, these gains are not significantly different (p > 0.05). We made sure to rectify the typographical and grammatical errors and the revised manuscript has been proofread by a native English speaker.

---

## [Decision Letter · Decision Letter 1]

6 Jan 2022

Deep learning model calibration for improving performance in class-imbalanced medical image classification tasks

PONE-D-21-32609R1

Dear Dr. Rajaraman,

We’re pleased to inform you that your manuscript has been judged scientifically suitable for publication and will be formally accepted for publication once it meets all outstanding technical requirements.

Kind regards,

Thippa Reddy Gadekallu

Academic Editor

PLOS ONE

Additional Editor Comments (optional):

Reviewers' comments:

Reviewer's Responses to Questions

**Comments to the Author**

1. If the authors have adequately addressed your comments raised in a previous round of review and you feel that this manuscript is now acceptable for publication, you may indicate that here to bypass the “Comments to the Author” section, enter your conflict of interest statement in the “Confidential to Editor” section, and submit your "Accept" recommendation.

Reviewer #1: All comments have been addressed

Reviewer #2: All comments have been addressed

2. Is the manuscript technically sound, and do the data support the conclusions?

Reviewer #1: Yes

Reviewer #2: (No Response)

3. Has the statistical analysis been performed appropriately and rigorously? 

Reviewer #1: Yes

Reviewer #2: (No Response)

4. Have the authors made all data underlying the findings in their manuscript fully available?

Reviewer #1: Yes

Reviewer #2: (No Response)

5. Is the manuscript presented in an intelligible fashion and written in standard English?

Reviewer #1: Yes

Reviewer #2: (No Response)

6. Review Comments to the Author

Reviewer #1: The authors have addressed all the concerns. The research work should be shared with the science community.

Reviewer #2: (No Response)

7. PLOS authors have the option to publish the peer review history of their article (what does this mean?). If published, this will include your full peer review and any attached files.

Reviewer #1: **Yes: **Sharnil Pandya

Reviewer #2: **Yes: **Dr. Kadiyala Ramana

---

## [Editor Report · Acceptance letter]

10 Jan 2022

PONE-D-21-32609R1 

Deep learning model calibration for improving performance in class-imbalanced medical image classification tasks 

Dear Dr. Rajaraman:

I'm pleased to inform you that your manuscript has been deemed suitable for publication in PLOS ONE. Congratulations! Your manuscript is now with our production department. 

Kind regards, 

on behalf of

Dr. Thippa Reddy Gadekallu 

Academic Editor

PLOS ONE